# Unexpected Inhibitory Effect of Octenidine Dihydrochloride on *Candida albicans* Filamentation by Impairing Ergosterol Biosynthesis and Disrupting Cell Membrane Integrity

**DOI:** 10.3390/antibiotics12121675

**Published:** 2023-11-28

**Authors:** Ting Fang, Juan Xiong, Li Wang, Zhe Feng, Sijin Hang, Jinhua Yu, Wanqian Li, Yanru Feng, Hui Lu, Yuanying Jiang

**Affiliations:** Department of Pharmacy, Shanghai Tenth People’s Hospital, School of Medicine, Tongji University, Shanghai 200072, China

**Keywords:** octenidine dihydrochloride, anti-filamentation, *Candida albicans*, ergosterol biosynthesis

## Abstract

*Candida albicans* filamentation plays a significant role in developing both mucosal and invasive candidiasis, making it a crucial virulence factor. Consequently, exploring and identifying inhibitors that impede fungal hyphal formation presents an intriguing approach toward antifungal strategies. In line with this anti-filamentation strategy, we conducted a comprehensive screening of a library of FDA-approved drugs to identify compounds that possess inhibitory properties against hyphal growth. The compound octenidine dihydrochloride (OCT) exhibits potent inhibition of hyphal growth in *C. albicans* across different hyphae-inducing media at concentrations below or equal to 3.125 μM. This remarkable inhibitory effect extends to biofilm formation and the disruption of mature biofilm. The mechanism underlying OCT’s inhibition of hyphal growth is likely attributed to its capacity to impede ergosterol biosynthesis and induce the generation of reactive oxygen species (ROS), compromising the integrity of the cell membrane. Furthermore, it has been observed that OCT demonstrates protective attributes against invasive candidiasis in *Galleria mellonella* larvae through its proficient eradication of *C. albicans* colonization in infected *G. mellonella* larvae by impeding hyphal formation. Although additional investigation is required to mitigate the toxicity of OCT in mammals, it possesses considerable promise as a potent filamentation inhibitor against invasive candidiasis.

## 1. Introduction

Invasive candidiasis has been proposed as a severe global health problem mainly caused by *C. albicans* [1,2]. Traditionally, antifungal drugs have antifungal activity by interfering with essential biological processes to kill fungi or inhibit fungal growth, such as azoles inhibiting ergosterol biosynthesis, polyenes directly targeting ergosterol in the plasma membrane, and echinocandins blocking cell wall biogenesis [3]. However, these antifungal drugs have many shortcomings that limit their clinical applications, such as severe side effects, limited antifungal spectra, and drug resistance [4]. Therefore, new antifungal agents are urgently needed to ameliorate the increased morbidity and mortality of invasive candidiasis. 

*C. albicans* is an opportunistic pathogen because *C. albicans* can exist in the host as a normal commensal. However, when the equilibrium tilts in favor of *C. albicans*, it will transform into pathogenic fungi by expressing virulence factors, such as a yeast-to-hyphae morphological transition, adhesion and invasion to the host, and biofilm formation [5]. Thus, targeting the virulence factors of *C. albicans* to reduce its virulence should be a potential antifungal strategy [6,7]. Filamentation is not only an important virulence factor for *C. albicans* in invasive candidiasis but also a critical event that enhances other *C. albicans* virulence factors during invasive candidiasis, such as adhesion, invasion, and escaping from phagocytes [8,9]. Some deletion mutants with defects in filamentation in *C. albicans* attenuate its virulence in vivo [10]. Thus, inhibiting hyphal transition may provide a way to attenuate the virulence of *C. albicans* specifically and is a promising therapeutic strategy for treating invasive candidiasis. However, few antifungal drugs inhibit yeast-to-hyphal conversion in *C. albicans*.

It is well known that developing new drugs is costly, including time and funding costs [11]. Repurposing existing Food and Drug Administration (FDA)-approved drugs is thus an attractive option for developing filamentation inhibitors because the costs associated with preclinical testing are avoided and the safety profiles and pharmacological characteristics are already established. In this study, to investigate the possibility of developing hyphal growth inhibitors, we screened a library of 2372 FDA-approved drugs in vitro to identify compounds capable of inhibiting hyphal growth in *C. albicans*. Of these FDA-approved drugs, octenidine dihydrochloride (OCT) was the most promising agent for inhibiting the filamentation of *C. albicans*.

## 2. Results

### 2.1. Identification of the Inhibitory Effect of OCT on C. albicans Filamentation

We conducted high-throughput screening assays utilizing a filamentous phenotype to evaluate the inhibitory activity of compounds from a library comprising 2372 FDA-approved drugs against hyphal growth. As a control hyphal growth inhibitor, we employed amphotericin B (AmB) at a concentration of 1.56 μM. Our findings revealed that, in a hyphae growth medium, RPMI 1640, 117 compounds exhibited comparable inhibitory effects on the filamentation of *C. albicans* to AmB at a concentration of 100 μM (Figure 1a). Twelve drugs with known antifungal properties, including AmB, econazole, econazole nitrate, isoconazole nitrate, miconazole nitrate, neticonazole, neticonazole hydrochloride, sulconazole mononitrate, tioconazole, anidulafungin, micafungin sodium, and flucytosine [12,13,14], were excluded from the study. The remaining 105 drugs were tested for their minimum hyphae-inhibiting concentration (MHIC) in RPMI 1640 media by serial 2-fold dilutions ranging from 100 μM to 0.0975 μM (Appendix A). It was observed that 14 out of the 105 drugs had MHIC values less than or equal to 12.5 μM (Figure 1a,c). To further verify the inhibitory effect of these 14 drugs on filamentation, we tested the MHIC values of these drugs in various hyphal growth-induced media, including RPMI1640, YPD + FBS (YPD with 10% fetal calf serum (FBS)), YPGlcNAc (YP with 5 mM N-acetyl-d-glucosamine (GlcNAc)), and Spider media. Broxyquinoline, clioquinol, chloroxine, and silver sulfadiazine exhibited inadequate inhibition of hyphal growth in *C. albicans* when tested in the YPD + FBS, YPGlcNAc, and Spider media as their MHIC values exceeded or were equal to 25 μM (Figure 1d). Similarly, MHIC values of tafenoquine succinate and thonzonium bromide exceeded or were equal to 25 μM against hyphal growth in *C. albicans* when tested in YPD + FBS and Spider media (Figure 1d). Benzethonium chloride, cetylpyridinium chloride, cetylpyridinium chloride monohydrate, and domiphen bromide failed to inhibit the hyphal growth of *C. albicans* in the Spider medium, and tavaborole failed in the YPD +FBS medium as their MHIC value reached up to 50 μM (Figure 1d). Likewise, visomitin had little inhibitory effect on the hyphal growth of *C. albicans* in the Spider medium, with an MHIC value of 25 μM (Figure 1d). In contrast, otilonium bromide exhibited a moderate inhibitory effect on the hyphal growth of *C. albicans* in various media, including RPMI1640, YPD+FBS, YPGlcNAc, and Spider, with MHIC values of 12.5 μM, 12.5 μM, 12.5 μM, and 6.25 μM, respectively (Figure 1d). Notably, OCT demonstrated the most potent inhibitory effect, with an MHIC value less than or equal to 3.125 μM across all four hyphal growth-induced media (Figure 1b,d,e). Furthermore, we found that OCT effectively inhibits the hyphal growth of 6 *C. albicans* ATCC strains and 12 *C. albicans* clinical isolates, with an MHIC value of 3.125 μM (Appendix A). These results suggest that OCT may be repurposed as an antifungal agent by inhibiting the hyphal growth of *C. albicans*.

### 2.2. OCT Inhibits Biofilm Formation and Disrupts Mature Biofilm

Fungal biofilms are frequently encountered in clinical infections and are considered a formidable challenge to have even over 1000-fold antifungal resistance [15]. The significance of hyphal morphology in biofilm formation and maintenance in *C. albicans* has been established [16]. Therefore, considering the inhibitory effect of OCT on filamentation in *C. albicans*, it was hypothesized that OCT could also inhibit biofilm formation. To investigate this conjecture, we utilized the 2,3-bis-(2-methoxy-4-nitro-5-sulfophenyl)-2H-tetrazolium-5-carboxanilide (XTT) assay to evaluate the inhibitory effects of OCT on biofilm formation and the disruptive effects of OCT on mature biofilm. XTT, acting as a substrate of mitochondrial dehydrogenase, can be reduced by viable cells to produce orange products that are soluble in water. When XTT is combined with the electron coupler phenazine methyl sulfate, the absorbance of the resultant water-soluble product is directly proportional to the number of living cells present, corresponding to the biofilm content. OCT at 4 μM inhibited biofilm formation by more than 92.54 ± 5.22% relative to the control (*p* < 0.0001) (Figure 2a), and its half-inhibiting concentration (IC_50_) was 2.63 ± 0.33 μM for inhibiting biofilm formation. Similarly, OCT showed an eradication effect on the mature biofilm of *C. albicans*, with an IC_50_ of 3.16 ± 0.06 μM. Specifically, OCT eradicated mature biofilm by 71.88 ± 9.18% at 4 μM relative to the control (*p* < 0.0001) and completely eradicated mature biofilm at 8 μM (Figure 2b).

### 2.3. OCT Has Potent Cytotoxicity and Acts as a Fungicidal Agent

Many small molecule compounds that inhibit filamentation also have cytotoxicity and inhibit fungal growth [9]. We wondered if OCT belongs to this situation whereby OCT has hyphal inhibition ability and cytotoxicity. We tested the minimum inhibitory concentration (MIC) values of OCT against 12 standard strains and 54 clinical strains of *Candida* species, including *C. albicans* (*n* = 19), *Candida auris* (*n* = 4), *Candida glabrata* (*n* = 8), *Candida tropical* (*n* = 12), *Candida parapsilosis* (*n* = 11), *Candida krusei* (*n* = 8), and *Candida guilliermondii* (*n* = 4). The MIC values of OCT against *C. albicans*, *C. auri*, *C. parapsilosis*, *C. tropical*, and *C. krusei* ranged from 0.39 µM to 0.78 µM (Figure 3a). 

The antifungal activity of OCT against *C. albicans* was further confirmed by an inhibition growth curve assay. OCT completely inhibited the growth of *C. albicans* at 2 µM compared to the control (*p* < 0.0001), but 2 µM fluconazole (FLC) could not (Figure 3b). The time-kill curve assay showed that OCT exhibited fungicidal effects at 1, 2, and 4 µM within 4 h against *C. albicans*. OCT at 1 µM and 2 µM decreased by 3.22 ± 0.07 and 5.32 ± 0.85 log_10_ CFU/mL, respectively, while at 4 µM, it completely killed the *C. albicans* cells (*p* < 0.001) (Figure 3c). We further measured the minimum fungicidal concentration (MFC) value of OCT, which was 1 μM in the RPMI 1640 medium, the same as the MIC of OCT, so it is a fungicidal drug (Figure 3d).

Furthermore, we exposed *C. albicans* with a cell density of 1 × 10^6^ cells/mL to varying concentrations of OCT (0, 1, 2 μM) in the RPMI 1640 medium for 4 h at a temperature of 37 °C. The morphological changes in *C. albicans* filamentation were observed using a microscope, and the growth of *C. albicans* was assessed by transferring them onto an SDA plate. The findings revealed that at an OCT concentration of 1 μM, which did not impede mycelial growth, the fungi exhibited viability compared to the control group. When the OCT concentration reached 2 μM, which was the concentration that completely inhibited mycelial growth, 99.08 ± 0.76% of the fungi died (Figure 3e). These results demonstrated that the concentration of cytotoxicity exerted by OCT and the concentration of inhibition of mycelium are similar and suggested that OCT may inhibit both the growth and the proliferation of *C. albicans* hyphae by interfering with a certain biological process.

### 2.4. The Antifungal Activity of OCT Depends on Intracellular Ergosterol

There is a presence of multiple homologous proteins in both *C. albicans* and *Saccharomyces cerevisiae*. Despite the incapability of *S. cerevisiae* to generate hyphae, the similar MIC values of OCT against *C. albicans* and the MHIC value of OCT against *C. albicans* filamentation indicate that the gene deletion library of *S. cerevisiae* can be employed to investigate the signaling pathways affected by OCT and thereby uncover its antifungal mechanism.

The activation of the cyclic adenosine monophosphate-protein kinase A (cAMP-PKA) pathway through Ras1 in serum triggers the activation of adenylyl cyclase Cyr1, causing the dissociation of Tpk1 and Tpk2 from Bcy1. This sequential signaling cascade ultimately leads to the phosphorylation of Efg1. The regulation of the yeast-to-hyphae morphological transition in response to temperature is primarily controlled by the heat shock protein 90 (Hsp90). Hsp90 inhibits filamentation under non-inducing conditions through the cAMP-PKA signaling pathway and other independent mechanisms [17,18]. When temperatures are elevated, the repression of Ras1 by Hsp90 is relieved, ultimately leading to the activation of the transcription factor Efg1 for hyphal induction, as described above [19]. In a manner independent of the cAMP-PKA pathway, Hms1 is recruited to the promoter of the *UME6* gene by involving cyclin-dependent kinase Pho85 and cyclin Pcl1 in response to elevated temperatures (Figure 4a) [18]. The antifungal activity of OCT is not affected by elevated temperatures (37 °C and 42 °C) compared to 30 °C (Figure 4b). Similarly, the presence of geldanamycin, an Hsp90 inhibitor, does not influence the antifungal activity of OCT (Figure 4c). Furthermore, we conducted susceptibility tests on the *ras1*Δ, *pde1*Δ, *tpk2*Δ (homologous to the *TPK1* gene in *C. albicans*), *sok2*Δ (homologous to the *EFG1* gene in *C. albicans*), and *ume6*Δ null mutants obtained from a *S. cerevisiae* gene deletion library. Our findings indicate that the absence of these genes does not impact the antifungal activity of OCT (Figure 4d). Collectively, the mechanism of Hsp90-mediated hyphal formation does not appear to be involved in the inhibitory effect of OCT on hyphal growth.

The regulation of pH-mediated hyphal morphogenesis is primarily governed by the transcriptional regulator Rim101 [20]. In acidic environments (pH < 6.5), Rim101 exists in its unprocessed, inactive, full-length state. However, in alkaline environments (pH ≥ 6.5), the inactive full-length Rim101 is recruited to the endosomal membrane through its C-terminal inhibitory domain binding to the scaffold protein Rim20. This interaction facilitates the proximity of Rim101 to Rim13, a protease resembling calpain. Subsequently, Rim13 cleaves the inhibitory domain located at the C-terminus of Rim101, thereby facilitating the translocation of the active form to the nucleus. Once in the nucleus, Rim101 regulates transcriptional responses dependent on a neutral-alkaline pH [21]. The activation of filamentation mediated by Rim101 is contingent upon Efg1. Moreover, the activation of Efg1 by Rim101 also triggers the activation of Tec1, which subsequently induces the expression of genes associated with filament formation (Figure 4a) [22]. Susceptibility tests were conducted on the *rim101*Δ, *rim13*Δ, and *tec1*Δ null mutants obtained from the *S. cerevisiae* gene deletion library. The absence of the *RIM101* gene resulted in an increased susceptibility of *S. cerevisiae* to OCT (Figure 4d). However, the susceptibility of *C. albicans* to OCT, including in MIC and spot assays, remained unaffected by the loss or over-expression of the *RIM101* gene (Figure 4e,f), indicating that OCT inhibits the hyphal growth of *C. albicans* independently of the Rim101-mediated hyphal growth mechanism.

As previously mentioned, the role of Efg1 in the induction of hyphal formation in *C. albicans* in response to serum, CO_2_, alkaline pH, and GlcNAC has been discussed. The transportation of GlcNAc into *C. albicans* is facilitated by the plasma membrane transporter known as Ngt1, which influences downstream factors via Ras1. In order to assess the impact of serum and GlcNAC on the antifungal activity of OCT, the MIC value of OCT was also examined and found to remain unchanged (Figure 4f).

The deprivation of nitrogen and amino acids serves as a stimulus for the activation of Mep2, leading to a series of interconnected processes that ultimately result in the activation of Cph1-dependent MAPK and cAMP-dependent signaling pathways. This sequence of events encompasses the activation of Cst20 and Ras1 through Mep2, subsequently triggering the activation of Hst7 and Cek1, culminating in the activation of Cph1 and the induction of hyphal formation (Figure 4a) [23]. Susceptibility tests were conducted on the *ste7*Δ (homologous to the *HST7* gene in *C. albicans*) and *fus3*Δ (homologous to the *CEK1* gene in *C. albicans*) null mutants obtained from an *S. cerevisiae* gene deletion library. The *ste7*Δ mutant exhibited increased sensitivity to OCT, while the *fus3*Δ mutant did not show any change in sensitivity to OCT (Figure 4d). A YNB medium was employed to simulate low nitrogen conditions and assess the sensitivity of *C. albicans* to OCT, and it was observed that the MIC value of OCT remained unchanged (Figure 4g). Similarly, the susceptibility of *C. albicans* to OCT, including in MIC and spot assays, remained unaffected by the loss or over-expression of the *HST7* gene (Figure 4e,f), indicating that OCT inhibits the hyphal growth of *C. albicans* independently of the nitrogen starvation-activated signaling pathway.

The inhibition of growth and the hyphal growth of *C. albicans* can be achieved by depleting its intracellular ergosterol content. Given that the antifungal activity and inhibiting hyphal growth effects of OCT are not reliant on conventional signaling pathways associated with mycelial growth, it is hypothesized that the proposed antifungal activity and inhibiting hyphal growth effects of OCT are contingent upon the reduction of intracellular ergosterol levels in *C. albicans*. In our study, it was observed that the MIC value of OCT exhibited an increase from 1 μM to 4 μM following the addition of 100 μM ergosterol, whereas supplementation with 100 μM cholesterol did not exert any influence on the MIC value of OCT. The spot assay further confirmed this outcome (Figure 4h,i). Additionally, we investigated the impact of ergosterol on the inhibition of mycelium by OCT in the RPMI 1640 medium. The initial concentration of OCT was set as 16 μM, and it was determined that the MHIC value of OCT increased from 2 μM to 16 μM upon supplementation with 100 μM ergosterol (Figure 4j). To further investigate the impact of ergosterol on the antifungal ability and inhibition of mycelium by OCT, it was observed that the depletion of specific genes within the ergosterol biosynthesis pathway (Figure 4k), such as *ERG3*, *ERG4*, and *ERG6* in *S. cerevisiae*, resulted in hypersensitivity to OCT (Figure 4l). Similarly, the loss of the *ERG3*, *ERG5*, *ERG24*, and *UPC2* genes increased the susceptibility of *C. albicans* to OCT (Figure 4m). These findings collectively suggest that the antifungal activity and inhibitory effects on hyphal growth of OCT are dependent on impaired ergosterol biosynthesis.

### 2.5. OCT Induces Reactive Oxygen Species (ROS) Generation in C. albicans

Depleting intracellular ergosterol and augmenting intracellular ROS levels in *C. albicans* are important for the fungicidal activities of AmB and miconazole [24]. Given that OCT has fungicidal activity that depends on the depletion of ergosterol, we speculated that OCT could generate ROS and lead to oxidative stress damage to *C. albicans*. We used a 2, 7-dichlorodi-in hydroacetate (DCFH-DA) dye to detect intracellular ROS levels in *C. albicans* quantitatively. When *C. albicans* was exposed to OCT for 4 h, concentrations of 1 μM, 2 μM, and 4 μM OCT were found to induce an increase in ROS accumulation. This increase in ROS accumulation was evident through the observed rise in fluorescence intensity, which was measured to be 1727 ± 167 initially and subsequently increased to 3597 ± 16 (*p* < 0.001), 4121 ± 150 (*p* < 0.0001), and 3922 ± 509 (*p* < 0.0001), respectively (Figure 5a). The MIC value of OCT was assessed to investigate the further impact of NAC on the fungicidal activity of OCT. It was observed that the MIC value of OCT increased from 1 μM to 2 μM upon the addition of 10 mM NAC (Figure 5b). Additionally, a spot assay was conducted to evaluate the survival of *C. albicans* in the presence of 10 mM NAC. The results of this assay demonstrated a significant increase in the survival rate of *C. albicans* upon the addition of 10 mM NAC (Figure 5c). It was observed that 2 μM OCT inhibited the growth of mycelia, and the presence of 10 mM NAC did not alter the minimum concentration required for mycelial inhibition (Figure 5d). The inhibitory mechanism of OCT on hyphal growth depends on the reduction of ergosterol rather than the generation of ROS.

### 2.6. OCT Disrupts the Membrane Integrity of C. albicans 

Based on the inhibitory effects of OCT on ergosterol biosynthesis and its ability to induce the generation of ROS, it was postulated that OCT disrupts the membrane integrity of *C. albicans*. A cell leakage experiment was conducted to evaluate the permeability of the cell membrane, wherein the release of intracellular components into the medium was measured as an indicator of cell permeability. Specifically, the leakage components primarily consisted of nucleotides and proteins, which absorb light at 260/280 nm. The results revealed that the release of intracellular substance proteins and nucleic acid from *C. albicans* cells occurred at a concentration of 8 μM OCT. The extracellular release of proteins and nucleic acid exhibited a significant increase of 5.65 ± 0.22 and 5.84 ± 0.11 times, respectively, compared to the control (Figure 6a). These findings suggest that OCT exerts its antifungal effect by inducing membrane destruction. Given the crucial role of ergosterol in fungal cell membranes and its antagonistic effect on OCT’s antifungal activity, it was postulated that ergosterol may serve as a protective agent for the plasma membrane against OCT-induced damage. It was observed that the presence of 100 μM ergosterol restored intracellular substance protein and nucleic acid leakage levels to the control state (Figure 6a). Furthermore, considering the antifungal activity of OCT as an NAC antagonist, it was hypothesized that NAC could potentially safeguard the plasma membrane against OCT-induced harm. This hypothesis was supported by the discovery that adding 10 mM NAC led to the return of intracellular substance protein and nucleic acid leakage levels to the control state (Figure 6b). 

### 2.7. OCT Exhibits Antifungal Efficacy in a Galleria Mellonella Infection Model 

We utilized a *G. mellonella* candidiasis model to evaluate the efficacy of OCT in vivo. The *G. mellonella* larvae were infected with 7.0 × 10^5^
*C. albicans* cells. Each experimental group consisted of ten larvae and their progression was observed for 10 days. On the 2nd day, eight larvae from the control group perished, while the remaining two succumbed on the 8th day, resulting in a median survival of 2 days for the control group. Conversely, the treatment group receiving 0.25 mg/kg OCT exhibited a survival rate of 40% (*p* < 0.05), with a median survival extended to 4.5 days. Furthermore, the survival rate of the treatment group receiving a 0.5 mg/kg dosage was 90% (*p* < 0.0001). It is worth mentioning that the administration of OCT at dosages of 1 mg/kg and 2 mg/kg resulted in complete protection against invasive candidiasis in *G. mellonella* (*p* < 0.0001), with the median survival of these control groups remaining undefined for a period exceeding 10 days. Conversely, administering FLC at a dosage of 2 mg/kg did not exhibit a significant protective effect on the survival rate compared to the control group, as its median survival was only 3 days (Figure 7a). The study employed the periodic Acid-siff staining (PAS staining) technique to understand better the development of the *C. albicans* infection in *G. mellonella*. Subsequently, the efficacy of OCT in inhibiting *C. albicans* filamentation was evaluated through pathological sections. The larvae were infected with *C. albicans* cells and treated with either 2 mg/kg FLC or 2 mg/kg OCT or no treatment. After a 24 h treatment period, the larvae were fixed, PAS stained, and subjected to image scanning to assess filamentation. *C. albicans* was observed in the tissue of both drug-treated and untreated larvae. However, a significant presence of mycelial cells was observed in the untreated larvae and the group treated with 2 mg/kg FLC, indicating widespread infection. Conversely, in the group treated with 2 mg/kg OCT, a substantial reduction in *C. albicans* colonization was observed, with only a limited number of yeast cells and fewer areas of infection (Figure 7b). These findings suggest that OCT protects against invasive candidiasis in *G. mellonella* larvae by effectively inhibiting hyphal formation and eradicating *C. albicans*.

## 3. Discussion

The morphological plasticity of *C. albicans* is widely recognized as a significant determinant of its virulence as the filamentation process plays a pivotal role in candidiasis progression. In mucosal candidiasis, the hyphal forms of *C. albicans* effectively infiltrate both epithelial and endothelial cells, leading to consequential tissue damage [25]. Similarly, in the case of invasive candidiasis, filamentation is crucial for *C. albicans* to evade phagocytes and the overall immune response [26]. Consequently, employing an anti-filamentation compound holds great promise for treating candidiasis. The compound filastatin was identified through a high-throughput phenotypic screening methodology. This compound demonstrated significant inhibition of the adhesion of *C. albicans* to epithelial cells, hindered the growth of hyphae, and effectively prevented the formation of biofilms. It is worth noting that filastatin displayed antifungal properties in both a nematode model of invasive candidiasis and an ex vivo mouse model of vulvovaginal candidiasis [27], thus suggesting the feasibility of employing the anti-filamentation strategy. The present study employed a drug repositioning strategy, screening a library of 2372 FDA-approved drugs to identify compounds capable of inhibiting hyphal growth in *C. albicans*. The findings of this study demonstrate the strong anti-filamentation properties of OCT, which effectively inhibits the growth of hyphae in *C. albicans* across various hyphae-induced media, including RPMI1640, YPD+FBS, YPGlcNAc, and Spider media, even at low concentrations. Additionally, OCT exhibits significant inhibition of biofilm formation and disruption of mature biofilms in *C. albicans*. These results suggest that OCT holds promise as a potential FDA-approved drug for combating fungal hyphal formation.

OCT, a member of the bipyridine class, is a commonly employed antiseptic agent and cationic surfactant [28]. Its molecular structure is comprised of two independent cationic rings, which are connected by a hydrocarbon chain. Compared to quaternary ammonium compounds, OCT exhibits reduced toxicity due to the absence of amide and ester constituents within its chemical composition [29]. Previous studies have reported that OCT demonstrates antifungal and antibiofilm properties [30,31,32,33,34,35]. The current study reveals that OCT effectively inhibits the growth of hyphae in *C. albicans*, preventing biofilm formation and disrupting mature biofilms. The MIC values of OCT against *Candida* species showed strong antifungal effects on both standard *Candida* and clinical isolates. Considering the increase of drug-resistant *Candida* in the clinic, OCT or its analogs have the potential to help alleviate clinical resistance problems. Additionally, OCT successfully eradicates *C. albicans* cells in the mycelial morphology of *G. mellonella* larvae and exhibits antifungal efficacy in a *G. mellonella* infection model. However, it is important to note that the intraperitoneal injection of OCT to treat invasive candidiasis in mice can result in fatality due to its toxicity. Therefore, further research should be conducted to reduce the in vivo toxicity of OCT.

Recent reports have indicated that the antibacterial properties of OCT are attributed to its physical interaction with the bacterial cell membrane. Specifically, OCT is observed to penetrate the hydrophobic fatty acyl chain region of the bilayer, resulting in the induction of complete lipid disorder and the subsequent rapid disruption of the cell membrane in *Escherichia coli* [36]. In the present study, we discovered that ergosterol effectively counteracts the inhibitory effects of OCT on the vegetative and hyphal growth of *C. albicans*. Furthermore, our findings demonstrate that deleting ergosterol biosynthesis-related genes enhances the susceptibility of *C. albicans* to OCT. Moreover, the presence of ergosterol has been found to hinder the detrimental impact of OCT on the cellular membrane of *C. albicans*. These findings suggest that, in contrast to its antibacterial properties that involve physical disruption of the bacterial cell membrane, OCT may exhibit antifungal effects by diminishing the ergosterol levels within *C. albicans* cells, consequently compromising the structural integrity of the fungal cell membrane. Nevertheless, additional investigations are required to elucidate the precise antifungal mechanism of OCT.

Numerous small molecules, including cell cycle inhibitors, histone deacetylase inhibitors, and rapamycin, have been documented as inhibitors of the yeast-to-hyphal transition in *C. albicans* [37]. This anti-virulence strategy effectively incapacitates *C. albicans*, rendering it incapable of causing infection and restoring its benign commensal state while impeding the emergence of fungal resistance. It is worth noting, however, that many of these small molecules hinder filamentation and impede fungal growth. OCT similarly falls into this category, as its MIC and MHIC values coincide. Further investigation is required to ascertain the roles of cytotoxicity and the inhibition of filamentation by OCT in conferring protection against candidiasis.

In conclusion, the FDA-approved drug OCT demonstrates significant inhibitory effects on the hyphal growth and biofilm formation of *C. albicans* in vitro. The mechanism by which OCT inhibits hyphal growth may be attributed to its ability to inhibit ergosterol biosynthesis and generate ROS, consequently disrupting the integrity of the cell membrane. Additionally, OCT exhibits protective properties against invasive candidiasis in *G. mellonella* larvae by effectively eliminating the colonization of *C. albicans* in infected organs by inhibiting hyphal formation. Even though further research is needed to reduce the toxicity of OCT in mammals, OCT has great potential as a filamentation inhibitor against invasive candidiasis.

## 4. Materials and Methods

### 4.1. Strains, Primers, Agents, and Cultural Conditions

All *C. albicans* strains and primers used in this work are listed in Appendix A. The strains were cultivated in a YPD medium (1% yeast extract, 2% dextrose, 2% bacteriological peptone) at a temperature of 30 °C and a speed of 200 rpm overnight unless otherwise specified. An SDA (1% bacteriological peptone, 4% dextrose, 2% agar) plate was used in the fungal culture experiment. For the induction of hyphal growth, we utilized various media, including RPMI 1640 (10.4 g/L RPMI-1640 (Sigma-Aldrich, St. Louis, MO, USA), 3.45% MOPS, 0.2% NaHCO_3_, pH 7.0), Spider media (1% yeast extract, 0.2% K_2_HPO_4_, 1% Mannitol, pH 7.0), YPGlcNAc (1% yeast extract, 5 mM GlcNAc, 2% bacteriological peptone), and a YPD + 10% (*v*/*v*) FBS medium (Sigma-Aldrich, St. Louis, MO, USA). The FDA-approved drug library obtained from MCE in Shanghai, China, consists of 2372 compounds dissolved in dimethylsulfoxide (DMSO) and stored in separate 96-well plates. The storage solutions for OCT (5 mM) from MCE in Shanghai, China; FLC (6.4 mg/mL); and AmB (6.4 mg/mL) from Aladdin in Shanghai, China, all utilized DMSO as the solvent, which was sourced from Sangon Biotech in Shanghai, China. Additionally, ergosterol (10 mM) from Sangon Biotech in Shanghai, China, used 50% Tween 80:50% ethanol as the solvent. NAC (500 mM) from Aladdin in Shanghai, China, used H_2_O as the solvent and was filtered by a 0.22 μm filter membrane.

### 4.2. Deletion and Over-Expression of Target Genes

A fusion PCR technique was employed to acquire homologous recombinant DNA fragments for disrupting target genes. These fragments consisted of 78 base pairs that exhibited homology to both the upstream and downstream flanking sequences of the target gene and selective auxotroph markers (*HIS1* or *ARG4*). Subsequently, the resulting fusion product was directly introduced into the *C. albicans* strain SN152 and subjected to selection on a synthetic medium supplemented with the required auxotrophic substances. This process facilitated the generation of a mutant lacking the target gene [38]. To achieve the over-expression of target genes, we substituted the inherent promoter of the target gene with a consistently over-expressed *ADH1* promoter, as previously described [39]. 

### 4.3. High-Throughput Screening Inhibitors of the Filamentation of C. albicans 

This study utilized 2372 compounds from the FDA-approved drug library to identify compounds that demonstrate inhibitory effects on the filamentation process of *C. albicans*. Initially, *C. albicans* cells were cultured overnight and diluted to a concentration of 5 × 10^5^ cells/mL using an RPMI 1640 medium. Subsequently, 99 μL of the *C. albicans* cell solution and 1 μL of the drug solution were added to each well of a 96-well plate, resulting in a working concentration of 100 μM for each drug. The *C. albicans* cells were then incubated for 4 h at 37 °C, allowing for the observation of the morphological transition from yeast to hyphae using an optical microscope (Motic, Hong Kong, China). We employed AmB (1.56 μM) as a control for non-hyphal growth. The 96-well plate was used to observe whether drugs inhibited hyphal growth in each well visually. Drugs that could not inhibit hyphal growth were eliminated and drugs that could inhibit hyphal growth were selected for the next round of screening.

### 4.4. Minimum Hyphae-Inhibiting Concentration (MHIC) Assay

The micro checkboard dilution method was utilized to determine the minimum concentration of the compound that inhibited the growth of hyphae. The MHIC concentration denoted the lowest concentration of the compound that prevented the visible growth of mycelium compared to control drugs. RPMI 1640, Spider, YPD + FBS, or YPGlcNac containing approximately 1 × 10^6^ cells/mL of *C. albicans* was dispensed at 50 μL per well in 96-well plates. Subsequently, 50 μL of compounds serially diluted from 200 μM to 0.195 μM were added to each well [40]. After cultivation, compounds with MHIC values equal to or less than 12.5 μM were chosen.

### 4.5. Biofilm Inhibition Assay

The biofilm inhibition assay was performed as described [41]. The XTT assay was utilized to quantify the density of the biofilm, and, subsequently, the percentage of biofilm inhibition was determined. To experiment, *C. albicans* overnight cultures were re-suspended in the RPMI 1640 medium at a concentration of approximately 1 × 10^6^ cells/mL. Subsequently, 100 µL of the cell suspension was added to each well of a 96-well plate. Incubation of the plates at 37 °C was conducted for 90 min for the anti-biofilm formation assay, while for the anti-mature biofilm assay, the incubation time was extended to 24 h. After incubation, the wells underwent a single wash with phosphate-buffered saline (PBS, pH 7.2 to 7.6) obtained from Sangon Biotech, Shanghai, China, to remove non-adherent cells. Subsequently, 100 μL of fresh RPMI 1640 medium, with or without OCT, was added in a serial dilution ranging from 64 μM to 0.125 μM. The plates were then incubated at 37 °C for 24 h. A dye solution was prepared by combining 9 mL of 0.5 mg/mL XTT with 1 mL of 0.32 mg/mL PMS (phenazine methyl sulfate) and allowed to stand for 15 min. Following incubation, the preceding medium in the 96-well plate was discarded, and the wells were gently rinsed three times with PBS. Subsequently, 100 μL of XTT-PMS staining solution was introduced into each well and incubated for 2 h in a lightless environment at 37 °C. Lastly, 75 μL of the supernatant from the 96-well plate was transferred to a fresh 96-well plate, and the optical density at a wavelength of 490 nm was determined using the Multiskan Sky instrument (ThermoFisher Scientific, Waltham, MA, USA).

### 4.6. Minimum Inhibitory Concentration (MIC) Assay

The antifungal activity of OCT was evaluated using the MIC assay following the guidelines provided by the Clinical and Laboratory Standards Institute (CLSI), M27-A3 [42]. The RPMI 1640 medium was employed, with approximately 2 × 10^3^ cells/mL of strains inoculated at 100 μL per well in flat-bottom, 96-well plates. Serial dilutions of OCT ranging from 100 μM to 0.391 μM were introduced into each well. *Candida* species cells were incubated at a temperature of 30 °C for 24 h, while *C. neoformans* and *C.gattii* cells were incubated at the same temperature for 72 h. The optical densities were subsequently measured using a Multiskan Sky (ThermoFisher Scientific, USA) at an absorbance of 600 nm. The MIC was defined as the lowest concentration of the compound that inhibited 50% or more of cell growth, as indicated by the OD_600nm_, compared to the control.

### 4.7. Dose-Matrix Titration Assays

Dose-matrix titration assays were utilized to evaluate the combined impacts of drugs [43]. Specifically, 50 μL of drug A at a concentration four times greater than the ultimate concentration was gradually dispensed, employing a two-fold serial dilution across seven plate columns. Subsequently, 50 μL of drug B at a concentration four times higher than the final concentration was gradually dispensed, employing a two-fold serial dilution along seven rows of the plate. In each well housing drugs, along with a single control well-lacking drugs, 100 μL of the RPMI 1640 medium was dispensed and inoculated with approximately 2 × 10^3^ cells/mL of strains and then incubated at 30 °C for 24 h.

### 4.8. Growth Inhibition Curve Assays

The growth inhibition curve assays were conducted following the methodology previously described [44]. In summary, a YPD medium was inoculated with 2 × 10^3^ cells/mL of *C. albicans* and dispensed at a volume of 100 μL per well into 96-well plates. Subsequently, 100 μL of solutions containing 2 μM OCT or 2 μM FLC were added to each well. The plates were then incubated with agitation at a temperature of 30 °C, and the optical density at 600 nm was measured at 15 min intervals for 48 h using a Tecan plate reader (Infinite 200 PRO, Grödig, Austria).

### 4.9. Time-Killing Curve Assays 

Experiments were conducted using sterile 50 mL culture tubes with a culture volume of 10 mL. Cell suspensions of SN152 were prepared by diluting an overnight culture in the RPMI 1640 medium to approximately 1 × 10^6^ cells/mL and then incubated with varying concentrations of OCT (0 μM, 1 μM, 2 μM, and 4 μM). At specific time intervals (0, 1, 2, 3, 4, 6, 8, 12, and 24 h), 100 μL of cell suspensions from 1:10 serial dilutions were applied to SDA plates for duplication. The number of clones on the plates was determined after incubation at 30 °C for 48 h [45]. Triplicate-sample aliquots were extracted, subjected to serial dilution, and plated on a suitable agar medium devoid of drugs. Following an incubation period of 48 h at a temperature of 30 °C, the number of viable colonies was quantified. The resulting time-killing curve was constructed, with the X-axis representing time and the Y-axis representing the logarithmic count (log_10_ cells per mL) of viable colonies.

### 4.10. Minimum Fungicidal Concentration (MFC) Assay

The MFC refers to the lowest antifungal agent concentration required to eliminate 99.9% of the initial inoculum [46]. To evaluate the MFC of OCT, the RPMI 1640 medium was utilized, and approximately 2 × 10^3^ cells/mL of strains were inoculated. The inoculum was dispensed to 100 μL per well in flat-bottom, 96-well plates. Subsequently, 100 μL of OCT was added to each well, with concentrations ranging from 32 μM to 0.0625 μM. After an incubation period of 24 h at 30 °C, the wells displaying turbidity near the predetermined threshold were transferred onto SDA plates. The growth of the transferred contents was then assessed after 48 h. The corresponding drug concentration was the MFC value when the plate grew aseptically.

### 4.11. Spot Assay

In the spot assay, *C. albicans* overnight cultures were subjected to a ten-fold dilution and, subsequently, yeast cells were placed as spots on YPD plates with or without agents. Each dot represents a five-fold dilution of yeast cells, with 10^7^, 10^6^, 10^5^, 10^4^, and 10^3^ cells/mL concentrations. The YPD plates were then incubated at 30 °C for 48 h and photographs were captured for inhibition analysis.

### 4.12. Cell Membrane Permeability Assay 

The cell membrane permeability of *C. albicans* cells was assessed following previously established methods [47]. In brief, *C. albicans* cells that had been cultured overnight were prepared in a 10 mL solution of PBS at a concentration of 2 × 10^7^ cells/mL. The cell suspension was then exposed to varying concentrations (1 μM, 2 μM, 4 μM, and 8 μM) of OCT and 2 μM of AmB, both with and without the addition of 100 μM ergosterol or 10 mM NAC for 4 h. A negative control consisting of untreated cells was included. Following the treatment period, the cell suspension was subjected to centrifugation at 10,000 rpm for 10 min, the resulting supernatant was filtered, and its absorbance was measured at 260/280 nm using a UV-vis spectrophotometer (ThermoFisher Scientific, Waltham, MA, USA).

### 4.13. Measurement of Reactive Oxygen Species (ROS) Production

The level of ROS accumulation was assessed using the 2′,7′-dichlorofluorescein diacetate (DCFH-DA) (Sigma-Aldrich, St. Louis, MO, USA) probe staining method [48]. *C. albicans* cells, cultured overnight, were diluted at a ratio of 1:100 and exposed to OCT at concentrations of 1 μM, 2 μM, and 4 μM for durations of 4 h at a temperature of 30 °C. Subsequently, the cells were rinsed and stained with 20 μg/mL DCFH-DA for 60 min without light. The overall fluorescence intensity of each sample was measured using a Tecan plate reader, with excitation and emission wavelengths set at 488 nm and 525 nm, respectively.

### 4.14. Evaluating the Antifungal Activity of OCT in a G. mellonella Infection Model

The efficacy of OCT was confirmed using the *G. mellonella* infection model [49]. *G. mellonella* larvae, obtained from Tianjin Huiyude Biotech Company (Tianjin, China), were selected based on an average weight of 300 mg and randomly assigned to five groups (*n* = 10 per group), with any larvae displaying signs of melanization being excluded. The larvae were infected with 5 μL of an SN152 suspension (7.0 × 10^5^ cells/larvae) using a Hamilton syringe and subsequently treated with a single injection of OCT (0.25, 0.5, 1, 2 mg/kg) or FLC (2 mg/kg). All *G. mellonella* larvae were incubated at 30 °C for 10 days. The mortality of *G. mellonella* was evaluated daily and subjected to statistical analysis using the Kaplan–Meier method, specifically employing the log-rank test. Following the survival investigation, the same larvae were utilized for histological examination. More specifically, after 24 h of infection and treatment, the larvae were immersed in a solution fixed with 4% paraformaldehyde. The obtained samples were then forwarded to Shanghai Borf Biotechnology Co. Ltd. for Periodic Acid-Schiff staining (PAS staining), and the resulting pathological sections were subsequently scanned and imaged.

### 4.15. Statistical Analysis

All experiments were performed in duplicate three times. The data were represented in grouped column plots with standard deviation error bars. Statistical data analyses were performed using ordinary one-way ANOVA analysis followed by Dunnett’s multiple comparisons test or two-way ANOVA analysis followed by Sidak’s multiple comparisons test. All analyses were conducted using GraphPad Prism 9.0 software. Statistical significance was set at one of the following *p*-values in the figures: * *p* < 0.05; ** *p* < 0.01; *** *p* < 0.001; **** *p* < 0.0001. Asterisks are used in figures to indicate the *p*-value.

## Figures and Tables

**Figure 1 antibiotics-12-01675-f001:**
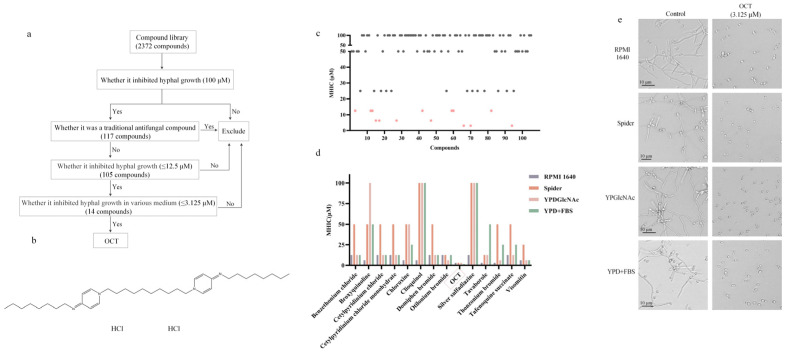
Identification of OCT for inhibiting hyphae activity. (**a**) Flow chart of screening an FDA-approved compound library (HY-L022, MCE^®^). (**b**) OCT chemical structure. (**c**) Minimum inhibitory hyphal concentrations of the screened 105 compounds in RMPI 1640 at 37 °C for 4 h. (**d**) Minimum inhibitory hyphal concentrations of the screened 14 compounds in four different hyphae-inducing mediums at 37 °C for 4 h. The red boxes indicate the final compound, Octenidine dihydrochloride (OCT). (**e**) OCT (3.125 μM) blocks *C. albicans* filamentation in response to diverse inducing cues, including RPMI 1640, Spider media, YPGlcNAc, and YPD + FBS at 37 °C for 4 h. Representative images are shown.

**Figure 2 antibiotics-12-01675-f002:**
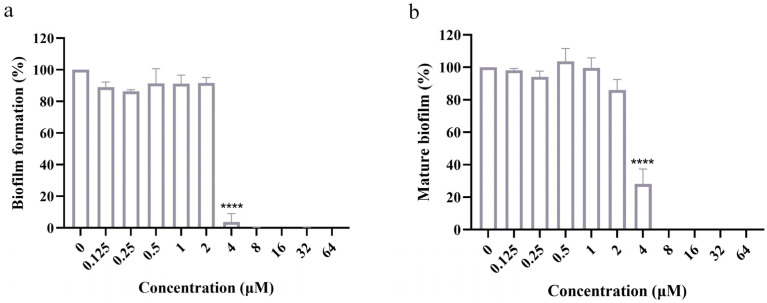
OCT inhibits biofilm formation and disrupts mature biofilm. (**a**) *C. albicans* was grown for 90 min and OCT (0-64 μM) was added for 24 h to assess the effects on biofilm formation by XTT reduction assay. (**b**) *C. albicans* was grown for 24 h and OCT (0–64 μM) was added for 24 h to assess the effects on biofilm maturation by XTT reduction assay. The significance of differences was determined by one-way ANOVA analysis followed by Dunnett’s multiple comparisons test (**** *p* < 0.0001).

**Figure 3 antibiotics-12-01675-f003:**
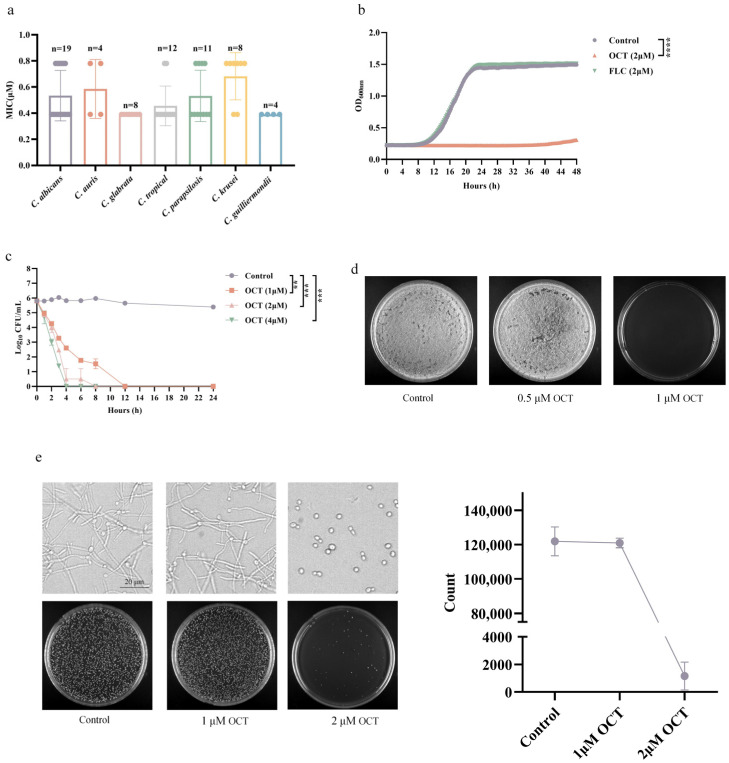
OCT has potent cytotoxicity and acts as a fungicidal agent. (**a**) MIC of OCT against *C. albicans* (*n* = 19), *Candida auris* (*n* = 4), *Candida glabrata* (*n* = 8), *Candida tropical* (*n* = 12), *Candida parapsilosis* (*n* = 11), *Candida krusei* (*n* = 8), and *Candida guilliermondii* (*n* = 4). (**b**) Growth curve of *C. albicans* cultured in the presence of 0 μM and 2 μM OCT and 2 μM FLC at 30 °C under shaking conditions. Growth was assessed by measuring OD_600nm_ every 15 min for 48 h. (**c**) The time-dependent effect of OCT on the growth curve of *C. albicans* was investigated by culturing the organism in the presence of varying concentrations of OCT (0 μM, 1 μM, 2 μM, and 4 μM) for 24 h at a temperature of 30 °C with shaking. At specific time points (0, 1, 2, 3, 4, 6, 8, 12, and 24 h), the resulting solution was applied onto SDA plates for duplication, and the number of colony-forming units (CFUs) was determined. Statistical analysis was performed using one-way ANOVA followed by Dunnett’s multiple comparisons test to assess the significance of differences. (** *p* < 0.01, *** *p* < 0.001, **** *p* < 0.0001). (**d**) Determination of the minimum fungicidal concentration (MFC) of OCT. (**e**) OCT treatment affects *C. albicans* viability, as indicated by the number of growing colonies on the plate on 0 μM, 1 μM, and 2 μM OCT treatment.

**Figure 4 antibiotics-12-01675-f004:**
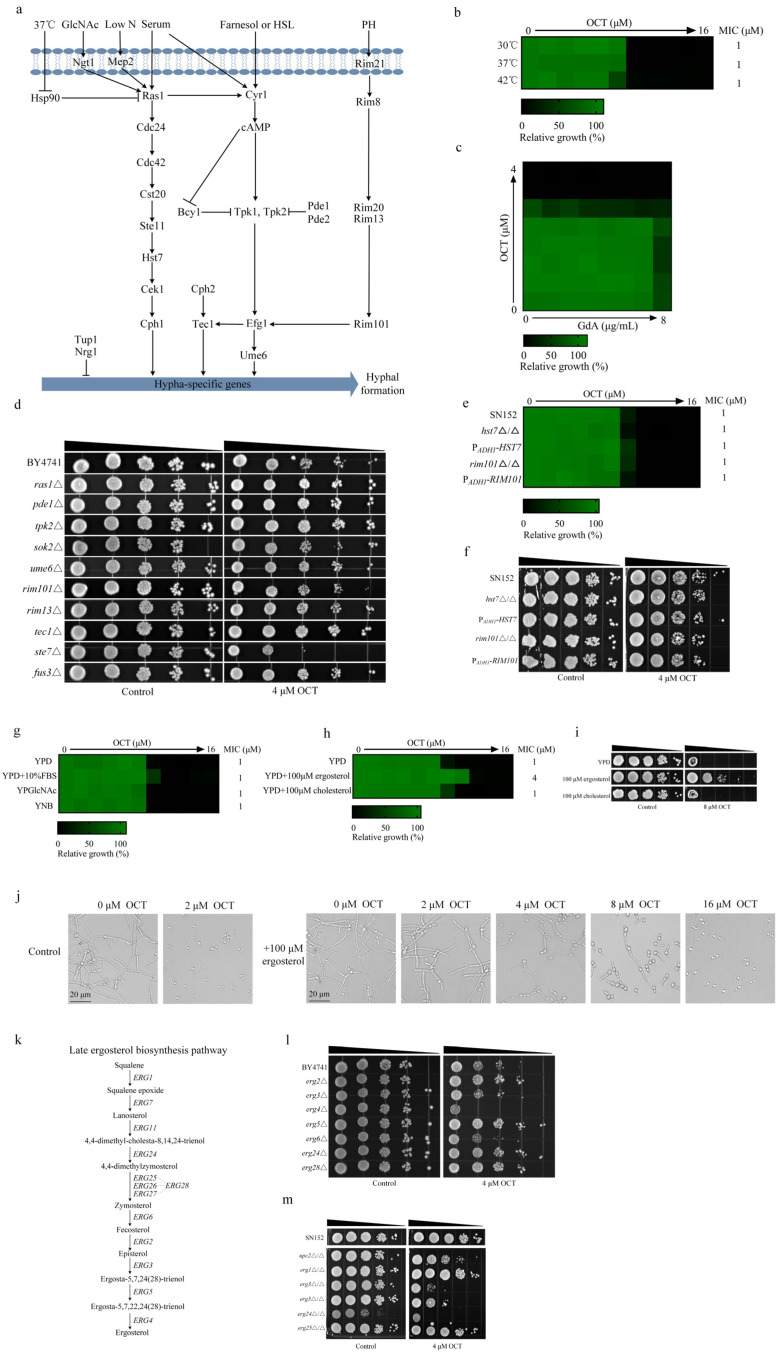
OCT depletes intracellular ergosterol. (**a**) The signaling pathways that regulate hyphal growth in response to environmental signals. (**b**) MIC assays of OCT were performed in YPD medium at 30 °C, 37 °C, and 42 °C against *C. albicans* (SN152) and growth was measured using absorbance at 600 nm after 24 h. Color quantitatively represents relative growth (see bar for scale at bottom). (**c**) Checkerboard assays were performed with OCT and GdA. Checkerboards were performed in YPD medium at 30 °C against *C. albicans* (SN152) and growth was measured using absorbance at 600 nm after 24 h. Color quantitatively represents relative growth (see bar for scale at bottom). (**d**) Spot assay of OCT against BY4741, *ras1*Δ, *pde1*Δ, *tpk2*Δ, *sok2*Δ, *ume6*Δ, *rim101*Δ, *rim13*Δ, *tec1*Δ, *ste7*Δ, and *fus3*Δ. Each dot represents a ten-fold dilution of yeast cells, with 10^7^, 10^6^, 10^5^, 10^4^, and 10^3^ cells/mL concentrations. The YPD plates were then incubated at 30 °C for 48 h before the plates were photographed. (**e**) MIC assays of OCT were performed in YPD medium at 30 °C against SN152, *hst7*Δ/Δ, *rim101*Δ/Δ, P*_ADH1_*-*HST7*, and P*_ADH1_*-*RIM101*. Assays were performed as in (**b**). (**f**) Spot assay of OCT against SN152, *hst7*Δ/Δ, *rim101*Δ/Δ, P*_ADH1_*-*HST7*, and P*_ADH1_*-*RIM101*. Assays were performed as in (**d**). (**g**) MIC assays of OCT were performed in YPD, YPD with 10% FBS, YPGlcNAc, and YNB medium at 30 °C against *C. albicans* (SN152). Assays were performed as in (**b**). (**h**) MIC assays of OCT were performed in YPD, YPD with 100 μM ergosterol, and YPD with 100 μM cholesterol medium at 30 °C against *C. albicans* (SN152). Assays were performed as in (**b**). (**i**) Spot assay of OCT against SN152 with or without 100 μM ergosterol or 100 μM cholesterol. Assays were performed as in (**d**). (**j**) OCT inhibits filamentation of SN152. Cells were grown at 37 °C for 4 h in the absence or presence of 100 µM ergosterol in RPMI 1640. Representative images are shown. (**k**) The signaling pathways that regulate ergosterol biosynthesis. (**l**) Spot assays of OCT against BY4741, *erg2*Δ, *erg3*Δ, *erg4*Δ, *erg5*Δ, *erg6*Δ, *erg24*Δ, and *erg28*Δ. Assays were performed as in (**d**). (**m**) Spot assays of OCT against SN152, *upc2*Δ/Δ, *erg1*Δ/Δ, *erg3*Δ/Δ, *erg24*Δ/Δ, and *erg25*Δ/Δ. Assays were performed as in (**d**).

**Figure 5 antibiotics-12-01675-f005:**
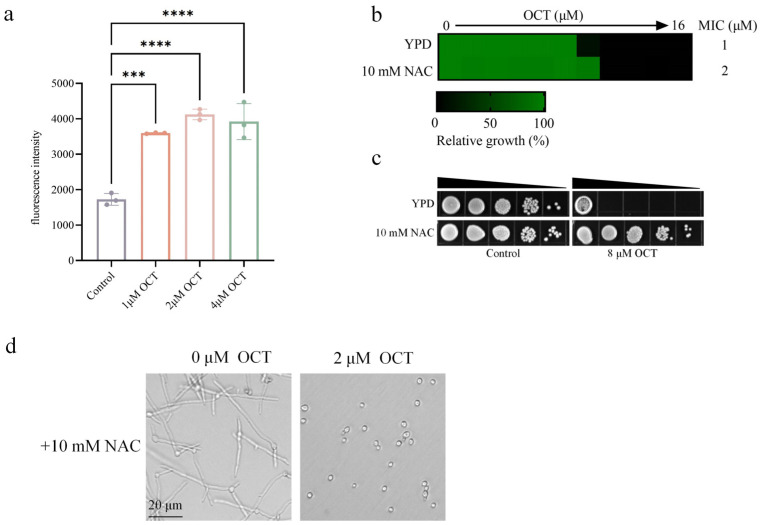
OCT disturbs the anti-oxidant system and influences the membrane integrity of *C. albicans*. (**a**). ROS levels after OCT treatment for 4 h with or without OCT. (**b**) MIC assays of OCT were performed in YPD and YPD with 10 mM NAC at 30 °C against *C. albicans* (SN152) and growth was measured using absorbance at 600 nm after 24 h. Color quantitatively represents relative growth (see bar for scale at bottom). (**c**) Spot assay of OCT against SN152 with or without 10 mM NAC. Each dot represents a ten-fold dilution of yeast cells, with 10^7^, 10^6^, 10^5^, 10^4^, and 10^3^ cells/mL concentrations. The YPD plates were then incubated at 30 °C for 48 h before the plates were photographed. (**d**) OCT inhibits filamentation of SN152. Cells were grown at 37 °C for 4 h in the presence or absence of 10 mM ergosterol in RPMI 1640. Representative images are shown. The significance of differences was determined by one-way ANOVA analysis followed by Dunnett’s multiple comparisons test (*** *p* < 0.001, **** *p* < 0.0001).

**Figure 6 antibiotics-12-01675-f006:**
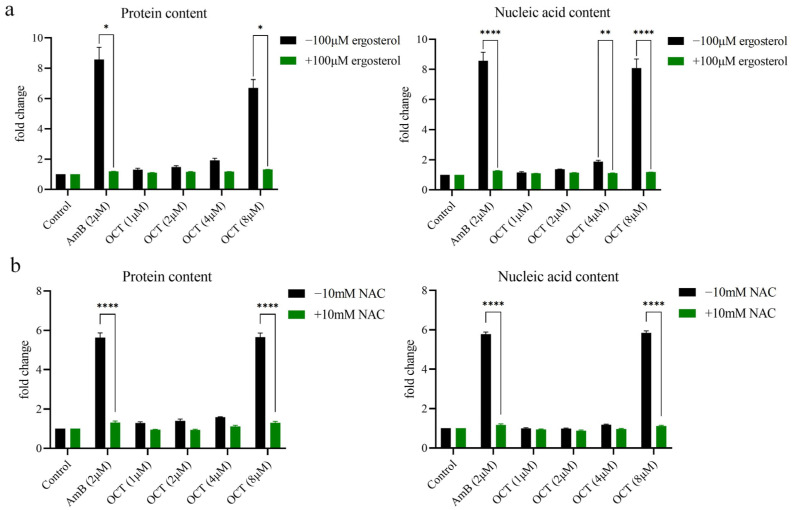
OCT disrupts the membrane integrity of *C. albicans*. (**a**) The soluble proteins and nucleic acid leakage in *C. albicans* after 0, 1, 2, 4, and 8 μM OCT and 2 μM AmB treatment with or without 100 μM ergosterol was assessed. (**b**) The soluble proteins and nucleic acid leakage in *C. albicans* were assessed after 0, 1, 2, 4, and 8 μM OCT and 2 μM AmB treatment with or without 10 mM NAC. The significance of differences was determined by two-way ANOVA analysis followed by Sidak’s multiple comparisons test (* *p* < 0.05, ** *p* < 0.01, **** *p* < 0.0001).

**Figure 7 antibiotics-12-01675-f007:**
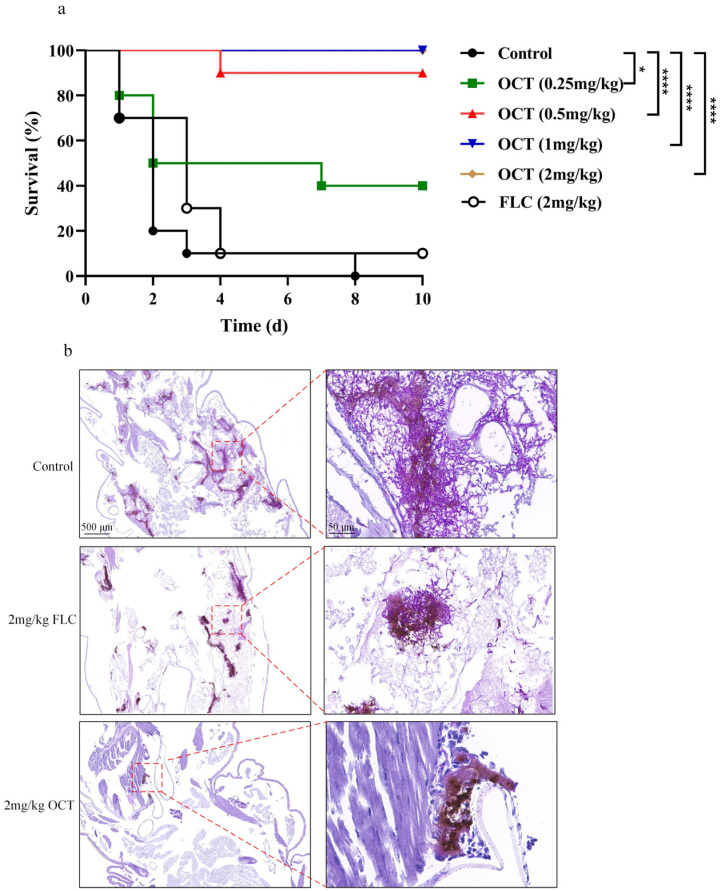
OCT exhibits antifungal efficacy in a *G. mellonella* infection model. (**a**) Survival curves of larvae infected with SN152 (7.0 × 10^5^ cells/larvae) and injected with 0.25 mg/kg, 0.5 mg/kg, 1 mg/kg, and 2 mg/kg doses of OCT or 2 mg/kg doses of FLC. Each curve represents a group of 10 larvae (*n* = 10) monitored daily for survival for up to 10 days after infection. (**b**) PAS staining of the larvae. The significance of differences was determined by the Kaplan-Meier method followed by the log-rank test (* *p* < 0.05, **** *p* < 0.0001).

## Data Availability

Data are contained within the article and Appendix A.

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
