# Peer review of "Unexpected Inhibitory Effect of Octenidine Dihydrochloride on Candida albicans Filamentation by Impairing Ergosterol Biosynthesis and Disrupting Cell Membrane Integrity"

_antibiotics, 2023, doi:10.3390/antibiotics12121675_

Round 1

Reviewer 1 Report

Comments and Suggestions for Authors

The manuscript reports in vitro and in vivo evaluation of octenidine dihydrochloride for the plausible treatment of invasive candidiasis, which may be achieved by the ability to inhibit ergosterol biosynthesis. I think it is interesting to the readers in this area of expertise.

Minor error: in Fig 7a, “FCZ” can be “FLC”.

Author Response

Response to comments of Reviewer #1

The manuscript reports in vitro and in vivo evaluation of octenidine dihydrochloride for the plausible treatment of invasive candidiasis, which may be achieved by the ability to inhibit ergosterol biosynthesis. I think it is interesting to the readers in this area of expertise.

  • We express our sincere gratitude for acknowledging our manuscript. We have diligently incorporated your valuable suggestions to enhance and refine our work.

Minor error: in Fig 7a, “FCZ” can be “FLC”.

  • Thank you for your helpful suggestion. We have corrected the “FCZ” into “FLC” in Figure 7a in the revised manuscript.

Reviewer 2 Report

Comments and Suggestions for Authors

In the present study the authors investigated the antifungal properties of octenidine dihydrochloride (OCT) against standard and clinical strains of Candida species. They found that OCT displays a remarkable inhibitory effect against hyphal growth and biofilm formation by C. albicans, and also disrupts mature biofilm. Further, the compound acts as a fungicidal agent against C. albicans and Candida non-albicans species. The mechanism underlying OCT's inhibition of hyphal growth is likely attributed to its ability to impede ergosterol biosynthesis and induce the generation of reactive oxygen species (ROS), compromising the integrity of the cell membrane. Additionally, OCT exhibits antifungal efficacy in Galleria mellonella infection model.

This is an interesting article that contributes to improve our knowledge of the antifungal properties of octenidine dihydrochloride. Although the antifungal effect of this compound against C. albicans and Candida non-albicans species has been already described in the literature, the mechanisms underlying the antifungal activity of OCT are not fully defined. However, in my opinion, the following issues need to be addressed before the article can be published:

 - Please clarify if Candida clinical isolates were tested for their susceptibility to antifungal drugs. It would be interesting to know whether OCT is efficacy against drug-resistant Candida strains. This is an important point that should be discussed.

  - Lines 106, 131 and 178, 293, 330 “Data represent two biological replicates”. It is not clear for me.  Are data the mean of two independent experiments? And each experiment was performed in duplicate or triplicate? Please clarify

   -Line 503” electron microscope” maybe” optical microscope”?

  -Line 535-536 “The antifungal activity of OCT was evaluated using the MIC assay following the  guidelines provided by the Clinical and Laboratory Standards Institute (CLSI) M27-A3” please add the reference.

  -Please update the bibliography by adding the following articles:

1. Chlorotaurine and Conventional Antiseptics against Candida spp. Isolated from Vulvovaginal Candidiasis. J Fungi (Basel). 2022 Jun 28;8(7):682. doi: 10.3390/jof8070682. PMID: 35887439; PMCID: PMC9322802.

2. Krasowski G, Junka A, Paleczny J, Czajkowska J, Makomaska-Szaroszyk E, Chodaczek G, Majkowski M, Migdał P, Fijałkowski K, Kowalska-Krochmal B, Bartoszewicz M. In Vitro Evaluation of Polihexanide, Octenidine and NaClO/HClO-Based Antiseptics against Biofilm Formed by Wound Pathogens. Membranes (Basel). 2021 Jan 17;11(1):62. doi: 10.3390/membranes11010062. PMID: 33477349; PMCID: PMC7830887.

3. Varghese VS, Uppin V, Bhat K, Pujar M, Hooli AB, Kurian N. Antimicrobial Efficacy of Octenidine Hydrochloride and Calcium Hydroxide with and Without a Carrier: A Broth Dilution Analysis. Contemp Clin Dent. 2018 Jan-Mar;9(1):72-76. doi: 10.4103/ccd.ccd_779_17. PMID: 29599588; PMCID: PMC5863414.

- Please, write in superscript the esponents.

- Candida species names should be italicized.

-  Minor English editing

Comments on the Quality of English Language

Minor editing of English language required

Author Response

Response to comments of Reviewer #2

This is an interesting article that contributes to improve our knowledge of the antifungal properties of octenidine dihydrochloride. Although the antifungal effect of this compound against C. albicans and Candida non-albicans species has been already described in the literature, the mechanisms underlying the antifungal activity of OCT are not fully defined. However, in my opinion, the following issues need to be addressed before the article can be published:

  • We express our sincere gratitude for acknowledging our manuscript. We have diligently incorporated your valuable suggestions to enhance and refine our work.

- Please clarify if Candida clinical isolates were tested for their susceptibility to antifungal drugs. It would be interesting to know whether OCT is efficacy against drug-resistant Candida strains. This is an important point that should be discussed.

  • Thanks for your constructive suggestions. We have tested the minimum inhibitory concentration (MIC) values of OCT against 12 standard strains and 54 clinical strains of Candida species, including albicans (n = 19), Candida auris (n = 4), Candida glabrata (n = 8), Candida tropical (n = 12), Candida parapsilosis (n = 11), Candida krusei (n = 8), Candida guilliermondii (n = 4). The MIC values of OCT against C. albicans, C. auri, C. parapsilosis, C. tropical, and C. krusei ranged from 0.39 µM to 0.78 µM (lines 136-141). We have further discussed the potent antifungal activity of OCT in lines 428-431.

  - Lines 106, 131 and 178, 293, 330 “Data represent two biological replicates”. It is not clear for me.  Are data the mean of two independent experiments? And each experiment was performed in duplicate or triplicate? Please clarify

  • We independently repeated each experiment three times and presented the results of one experiment as a representative. We deleted the unclear description “Data represent two biological replicates” in lines 105, 131, 176, 293, and 328 and added the description of the number of repetitions of the experiment in line 627.

   -Line 503” electron microscope” maybe” optical microscope”?

  • We have corrected the “electron microscope” into “optical microscope” in Line 502.

  -Line 535-536 “The antifungal activity of OCT was evaluated using the MIC assay following the guidelines provided by the Clinical and Laboratory Standards Institute (CLSI) M27-A3” please add the reference.

  • Thank you for your helpful suggestion. We have added references in line 539. (Reference: CLSI. Reference method for broth dilution antifungal susceptibility testing of yeasts. 4th ed. CLSI standard M27. Wayne, PA: Clinical and Laboratory Standards Institute; 2017.)

  -Please update the bibliography by adding the following articles:

  • We appreciate your helpful suggestion. We have cited the following references in line 422 and added these references in lines 720-726.
  1. Chlorotaurine and Conventional Antiseptics against Candida spp. Isolated from Vulvovaginal Candidiasis. J Fungi (Basel). 2022 Jun 28;8(7):682. doi: 10.3390/jof8070682. PMID: 35887439; PMCID: PMC9322802.
  2. Krasowski G, Junka A, Paleczny J, Czajkowska J, Makomaska-Szaroszyk E, Chodaczek G, Majkowski M, Migdał P, Fijałkowski K, Kowalska-Krochmal B, Bartoszewicz M. In Vitro Evaluation of Polihexanide, Octenidine and NaClO/HClO-Based Antiseptics against Biofilm Formed by Wound Pathogens. Membranes (Basel). 2021 Jan 17;11(1):62. doi: 10.3390/membranes11010062. PMID: 33477349; PMCID: PMC7830887.
  3. Varghese VS, Uppin V, Bhat K, Pujar M, Hooli AB, Kurian N. Antimicrobial Efficacy of Octenidine Hydrochloride and Calcium Hydroxide with and Without a Carrier: A Broth Dilution Analysis. Contemp Clin Dent. 2018 Jan-Mar;9(1):72-76. doi: 10.4103/ccd.ccd_779_17. PMID: 29599588; PMCID: PMC5863414.

- Please, write in superscript the esponents.

  • The manuscript has been meticulously examined, and we have corrected superscripts.

- Candida species names should be italicized.

  • We have examined the manuscript carefully to correct the Candida species names in italic format.

-  Minor English editing

  • We have examined the manuscript carefully to correct grammatical errors in English.

Reviewer 3 Report

Comments and Suggestions for Authors

The manuscript by Ting Fang and colleagues presents a very interesting and extremely well conducted study to investigate the anti-filamentation and cytotoxic properties of octenidine dihydrochloride towards C. albicans. A thorough investigation is also conducted into the compounds' mechanism of action.

Several minor points need to be addressed before publication:

-The quality of Figure 1 should be improved, and the bulletpoit in caption should be amended. Morevorer, the microscopy approach used for the acquisition of panel E images should be reported.

-Please carefully check the use of apex/subscript characters throughout the manuscript.

-Lines 184-187 This sentence is not clear, please rephrase.

-In table 2s the species of the strains with specific deleted genes should be reported

-Lines 501-503 Please detail how the use of electron microscopy can be applied to HTS of compounds

Author Response

Response to comments of Reviewer #3

The manuscript by Ting Fang and colleagues presents a very interesting and extremely well conducted study to investigate the anti-filamentation and cytotoxic properties of octenidine dihydrochloride towards C. albicans. A thorough investigation is also conducted into the compounds' mechanism of action.

Several minor points need to be addressed before publication:

  • We appreciate your positive comments. We have modified our manuscript according to your suggestion.

-The quality of Figure 1 should be improved, and the bulletpoit in caption should be amended. Morevorer, the microscopy approach used for the acquisition of panel E images should be reported.

  • Thank you for your helpful suggestions. We have improved Figure 1 and added a caption for Figure 1e in lines 103-105.

-Please carefully check the use of apex/subscript characters throughout the manuscript.

  • We have examined the manuscript carefully to correct the errors of apex/subscript characters.

-Lines 184-187 This sentence is not clear, please rephrase.

  • We have rephrased the sentence in lines 178-183.

-In table 2s the species of the strains with specific deleted genes should be reported

  • The mutant strains listed in Table S2 were all constructed in albicans. We have changed the name of TableS2 to 'C. albicans mutants were used in this study' in line 464.

-Lines 501-503 Please detail how the use of electron microscopy can be applied to HTS of compounds

  • First of all, we used an optical microscope rather than an electron microscope, which we have corrected in line 499 in the revised manuscript. Secondly, we use visual screening to high-throughput screening of compounds in 96-well plates. To be specific, we employed AmB (1.56 μM) as control for non-hyphal growth. The 96-well plate was used to visually observe whether drugs inhibited hyphal growth in each well. Drugs that could not inhibit hyphal growth were eliminated, and drugs that could inhibit hyphal growth were selected for the next round of screening. We have added this part in lines 499-503.